# Position: Graph Condensation Needs a Reset—Move Beyond Full-dataset Training and Model-Dependence

**Mridul Gupta** [* 1]  **Samyak Jain** [* 2]  **Vansh Ramani** [2]  **Hariprasad Kodamana** [3 1 4]  **Sayan Ranu** [2 1]

## Abstract

Graph Neural Networks (GNNs) are powerful tools for learning from graph-structured data, but their scalability is increasingly strained by the size of real-world graphs in domains like recommender systems, fraud detection, and molecular biology. Graph condensation—the task of generating a smaller synthetic graph that retains the performance of models trained on the original—has emerged as a promising solution. However, the dominant approach of gradient matching introduces a fundamental contradiction: it requires training on the full dataset to create the compressed version, thereby undermining the goal of efficiency. Worse still, these methods suffer from high computational overhead, poor generalization across GNN architectures, and brittle reliance on specific model configurations. Equally concerning is the community's reliance on misleading evaluation protocols such as node compression ratios, which fail to reflect true resource savings, condensation overhead, and illusory application to neural architecture search. These shortcomings are not incidental—they are systemic, and they obstruct meaningful progress.

In this position paper, we argue that graph condensation, in its current form, needs a reset. We call for moving beyond full-dataset training and model-dependent design, and instead advocate for methods that are lightweight, architecture-agnostic, and practically deployable.

[1]Yardi School of Artificial Intelligence, IIT Delhi, India [2]Department of Computer Science and Engineering, IIT Delhi, India [3]Department of Chemical Engineering, IIT Delhi, India [4]Indian Institute of Technology Delhi, Abu Dhabi, Zayed City, Abu Dhabi, UAE. Correspondence to: Samyak Jain <samyakjain1729@gmail.com>, Mridul Gupta <mridul.gupta@scai.iitd.ac.in>, Vansh Ramani <cs5230804@iitd.ac.in>, Hariprasad Kodamana <kodamana@iitd.ac.in>, Sayan Ranu <sayanranu@cse.iitd.ac.in>.

*Proceedings of the 43$^{rd}$ International Conference on Machine Learning*, Seoul, South Korea. PMLR 306, 2026. Copyright 2026 by the author(s).

By identifying key methodological flaws and outlining concrete research directions, we aim to reorient the field toward approaches that deliver on the true promise of condensation: efficient, generalizable, and usable GNN training at scale.

## 1. Introduction

Graph Neural Networks (GNNs) have emerged as the dominant paradigm for representation learning on graph-structured data(Veličković et al., 2018; Kipf & Welling, 2017; Hamilton et al., 2017; Xu et al., 2019; Nishad et al., 2021; Gupta et al., 2023b). Their utility has been demonstrated across diverse domains, including drug and material discovery (Xiong et al., 2021; Goyal et al., 2020; Bihani et al., 2023), traffic forecasting (Gupta et al., 2023a; Jain et al., 2021), recommendation systems (Gupta et al., 2025a; Sirohi et al., 2024), and modeling physical interacting systems (Bishnoi et al., 2024; 2023; Bhattoo et al., 2022), all of which share a common characteristic: the relational structure among entities is as informative as the entities' own features.

Training GNNs is costly, with runtime and memory per epoch growing with the number of nodes, edges, and feature dimensions in the graph (Hu et al., 2021). This cost becomes prohibitive on massive real-world graphs such as MAG240M (Hu et al., 2021) and large-scale workload datasets (Miao et al., 2021), which contain hundreds of millions of nodes and billions of edges. Compounding this challenge, training a GNN is rarely a one-time effort—it often involves multiple iterations to explore architectural choices, tune hyperparameters, and select appropriate loss functions. Such repeated training quickly becomes impractical at this scale. The limitations are equally pronounced in resource-constrained environments such as edge devices, which typically have $\leq 1$ GB RAM and no GPU support, making it infeasible to run even a single full-batch training epoch on moderately sized graphs.

Scalability techniques such as node sampling (Hamilton et al., 2017; Chen et al., 2018), graph partitioning (Liao et al., 2018; Chiang et al., 2019; Zeng et al., 2020), and low-rank embedding (Ding et al., 2022; Yang et al., 2023b)

aim to reduce resource demands, but they often struggle to simultaneously preserve global structural information, minimize dataset size, and ensure efficient computation. Sampling-based training, though efficient in theory, also becomes prohibitively expensive at scale (Liu et al., 2022b). Graph condensation (Jin et al., 2022b;a; Gao & Wu, 2023; Yang et al., 2023a; Fang et al., 2024; Zhang et al., 2024b; Gao et al., 2024a) offers a more promising direction: it seeks to synthesize a smaller graph $\mathcal{G}_c = (\mathcal{V}_c, \mathcal{E}_c, \mathbf{X}_c)$ such that a model trained on $\mathcal{G}_c$ achieves predictive performance comparable to that of a model trained on the original graph $\mathcal{G} = (\mathcal{V}, \mathcal{E}, \mathbf{X})$.

The seminal work GCOND (Jin et al., 2022b) was the first to formally define the graph condensation problem and propose a formal framework for it. Specifically, GCOND formulates condensation as a bi-level optimization task: it seeks to synthesize a smaller graph such that a GNN trained on it closely mimics the gradient trajectory of a GNN trained on the full graph. This formulation has strongly influenced subsequent research, with many follow-up methods adopting gradient-matching as a core design principle (Jin et al., 2022a; Gao & Wu, 2023; Mao et al., 2023; Li et al., 2023; Pan et al., 2023; Fang et al., 2024; Gao et al., 2024a; Liu et al., 2024c; Yang et al., 2023a). A closely related paradigm is to use expert parameter trajectory during training as the matching target inspired by (Cazenavette et al., 2022) and has been adapted in graph condensation domain by a few works (Zhang et al., 2024b; Zheng et al., 2024; Zhang et al., 2024a).

While the formulation of GCOND was inspired by distillation algorithms from the vision community, the field has diverged: vision increasingly uses VLM/LLM semantic priors (Gu et al., 2026; Zou et al., 2025) and diffusion-based distribution modeling (Gu et al., 2024; Su et al., 2024; Moser et al., 2024), whereas graph condensation must synthesize both structure and features while preserving message passing semantics and graph statistics.

The graph condensation landscape has matured to the point of attracting several benchmarking efforts (Gao et al., 2025b; Hashemi et al., 2024; Xu et al., 2024). While these studies provide valuable comparisons of existing methods, they largely refrain from questioning the foundational assumptions of the field. In this position paper, we argue that this reliance on gradient-matching (or on expert trajectory matching) fundamentally compromises the core goals of graph condensation—namely, scalability, model-agnosticism, and deployment feasibility. Our position is motivated by following key observations:

1. **Full-training paradox.** Gradient matching (or matching its training byproducts such as model weights) requires training on the full dataset to compute target trajectories, contradicting the very premise of condensation. Specifically, since training on the full dataset is a subroutine to

condensation, the condensation process is guaranteed to be more expensive than full-graph training itself.

2. **The NAS illusion.** In the absence of the core application to scale GNN training, a commonly cited alternative use-case for gradient-based condensation (and expert-trajectory matching based condensation) is neural architecture search (NAS). However, this justification is limited in scope: gradients are themselves functions of the model architecture, loss function, and optimization settings, which implies condensed graphs learned under one configuration primarily reflect that specific training dynamics. As such, there is little theoretical grounding and only mixed empirical evidence to support the claim that condensed graphs generated for one architecture can guide architecture search effectively for others. In particular, the dependence of gradient-based condensation on a fixed model configuration can restrict its effectiveness in cross-architecture settings, thereby limiting its utility as a general-purpose surrogate for architecture search (Table 4).

3. **Memory bottlenecks.** Methods based on gradient matching or expert trajectory matching often instantiate dense $N^2$ adjacency matrices. This makes even a one-time condensation infeasible on massive graphs such as MAG240M (Hu et al., 2021), let alone repeated condensation for iterative tuning or edge deployment (Table 2).

4. **Hidden costs and misleading metrics.** Current evaluation protocols overwhelmingly rely on node-count ratios to report the size of the condensed graph (Jin et al., 2022b;a; Zhang et al., 2024a; Feng et al., 2023; Mao et al., 2023; Li et al., 2023; Liu et al., 2024c; 2022a; Wang et al., 2024a; Xu et al., 2023; Liu et al., 2024a), but this single-factor metric hides critical details. It fails to account for edge density, feature dimensionality, or the resulting memory and compute costs—factors that directly impact practical feasibility. For instance, two condensed graphs with identical node counts may differ significantly in storage footprint and runtime depending on how densely connected or feature-rich they are. We argue that the field needs more comprehensive evaluation metrics that holistically capture all key resource demands and align better with the end-goals of graph condensation.

### 1.1. Contributions

The above pitfalls motivate a fundamental rethinking of the field, leading us to the following position: **Graph Condensation Needs a Reset: Move Beyond Full-dataset Training and Model-Dependence.** This reliance not only contradicts the foundational goals of condensation—scalability, model-agnosticism, and efficient deployment—but also undermines its broader applicability. In this position paper, we take concrete steps towards a course correction, guided by

the following key contributions:

1. **A principled reformulation of graph condensation.** In Section 2, we introduce a robust and generalizable definition of graph condensation that decouples the process from full-dataset training. Our formulation emphasizes three key dimensions: (1) *size efficiency*—capturing total resource footprint including nodes, edges, and features; (2) *task preservation*—retaining predictive utility across relevant tasks; and (3) *cross-architecture generalization*—ensuring performance does not hinge on a specific GNN configuration. This reformulation addresses the shortcomings of prior definitions (Jin et al., 2022b), which tightly bind the condensed graph to a particular model.

2. **Guidelines for fair and holistic comparison.** To structure benchmarking practices, we outline a new framework for analyzing condensation techniques along dimensions that matter in practice. This includes evaluating generalization to unseen architectures, robustness to hyperparameter changes, and actual resource consumption under deployment settings. By shifting the focus from abstract metrics to tangible outcomes, we aim to realign research priorities toward real-world feasibility.

3. **A call to action.** This paper aims to serve as a clear guide for best practices in graph condensation research. We invite the community to adopt this broader lens—one that values model-agnosticism, efficiency, and real-world deployability over narrowly defined optimization success. By doing so, we believe graph condensation can move beyond toy tasks and serve as a practical tool for modern graph learning at scale.

## 2. Graph Condensation: (Re)Formulation and Preliminaries

**Definition 2.1** (Graph). *A graph is a mathematical structure represented as $\mathcal{G} = (\mathcal{V}, \mathcal{E}, \mathcal{X})$, where $\mathcal{V}$ is a finite set of vertices (or nodes), $\mathcal{E} \subseteq \mathcal{V} \times \mathcal{V}$ is a set of edges (or arcs) that define relationships or connections between pairs of vertices, and $\mathcal{X} \in \mathbb{R}^{|\mathcal{V}| \times d}$ is a feature matrix, where each row $\mathbf{x}_v \in \mathbb{R}^d$ represents d-dimensional feature vector associated with the node $v \in \mathcal{V}$.*

For a target task $\zeta$ with desired output $\mathcal{Y}$ (e.g. node labels or graph labels) on $\mathcal{G}$, graph condensation seeks a compact surrogate graph $\mathcal{G}_c = (\mathcal{V}_c, \mathcal{E}_c, \mathcal{X}_c)$, where $\mathcal{V}_c$ is a condensed set of vertices, $\mathcal{E}_c \subseteq \mathcal{V}_c \times \mathcal{V}_c$ is a set of edges, and $\mathcal{X}_c \in \mathbb{R}^{|\mathcal{V}_c| \times d}$ is a feature matrix, with synthetic labels $\mathcal{Y}_c$, such that training a model $m$ on $\mathcal{G}_c$ preserves the predictive behavior of $m$ on the original graph $\mathcal{G}$. For brevity, we will henceforth couple each graph and its labels—writing $(\mathcal{G}, \mathcal{Y})$ simply as $\mathcal{G}$ and $(\mathcal{G}_c, \mathcal{Y}_c)$ as $\mathcal{G}_c$. We now formalize this objective by specifying the necessary conditions that a condensation algorithm must satisfy.

### 2.1. Necessary Conditions

We require any valid graph-condensation method to satisfy four core properties:

1. **$\epsilon$-task preservation:** For a target task, a model trained on $\mathcal{G}_c$ must achieve a desired performance ($\mathrm{performance}(.)$ is used to represent this abstractly, e.g. it may be accuracy, roc auc, etc.) not less than $\epsilon \geq 0$ of its performance on $\mathcal{G}$:

$$\mathrm{performance}(\mathcal{G}_c) \geq \mathrm{performance}(\mathcal{G}) - \epsilon.$$

2. **Compactness:** The goal of graph condensation is to produce a significantly smaller graph $\mathcal{G}_c$ that preserves the predictive utility of the original graph $\mathcal{G}$, i.e., $|\mathcal{G}_c| \ll |\mathcal{G}|$, according to an appropriate size metric. Two such metrics are commonly used in the literature:

   - **Byte-size** (Gupta et al., 2025b; 2024): This metric captures the total memory footprint of a graph $\mathcal{G}$, assuming float-encoded node features and integer-encoded edges. For a graph with feature dimension $d$, the byte-size is defined as:

   $$|\mathcal{G}| = d \cdot |\mathcal{V}| \cdot s_f + 2 \cdot |\mathcal{E}| \cdot s_i,$$

   where $s_f$ and $s_i$ denote the byte sizes of float and integer types respectively (e.g., $s_f = 2$, $s_i = 1$). This metric accounts for both feature and edge storage, making it more comprehensive than simpler alternatives. We further argue that since the complexity of message-passing is $O(|\mathcal{V}| \cdot d^2 + |\mathcal{E}| \cdot d) = O(\frac{1}{2s_i}d(d \cdot |\mathcal{V}| \cdot 2s_i + 2|\mathcal{E}| \cdot s_i)) = O(d \cdot |\mathcal{G}|)$ when $s_f = 2s_i$ (a common setting in practice), the byte-size metric is proportional to the computational cost of the forward pass in message-passing neural networks, making it a practical and faithful measure of graph compactness.

   - **Condensation ratio** (Jin et al., 2022b; Liu et al., 2024a; Zheng et al., 2024; Wang et al., 2024b): Defined as the node-count ratio $|\mathcal{V}'|/|\mathcal{V}|$, this metric is widely used due to its simplicity. However, it ignores edge density and feature dimensionality, and can therefore be misleading when evaluating the actual resource savings achieved through condensation.

3. **Generalization:** Retraining on $\mathcal{G}_c$ under reasonable changes in model architecture, model initializations, or hyperparameters yields performance comparable to training on $\mathcal{G}$.

4. **Compute efficiency:** The total cost of condensation plus training on $\mathcal{G}_c$ must not exceed the cost of training on $\mathcal{G}$:

$$C_{\mathrm{cond}}(\mathcal{G}) + C_{\mathrm{train}}(\mathcal{G}_c) \leq C_{\mathrm{train}}(\mathcal{G}).$$

Here cost ($C$) must be understood as running time, as in total CPU hours plus total GPU hours, appropriately weighted by their power consumption.

To encapsulate the above desiderata, we propose the following reformulation of the problem.

**Definition 2.2** (Size-Constrained Graph Condensation)**.** *Let $\mathcal{G}$ be the original graph, $\mathcal{M}$ be the class of GNNs over which we want to generalize (e.g. message-passing GNNs), $B$ a budget in bytes, and $\delta \in (0, 1)$ a confidence level. We seek the smallest performance gap $\epsilon$ such that the condensed graph $\mathcal{G}_c$, under the constraint $|\mathcal{G}_c| \leq B$, and $m(\cdot)$ is the output of the model $m$ from class $\mathcal{M}$:*

$$(\mathcal{G}_c^*, \epsilon^*) = \arg\min_{\mathcal{G}', \epsilon} \epsilon \quad \text{s.t.}$$

$$|\mathcal{G}'| \leq B, \text{ and } \Pr_{m \sim \mathcal{M}}\big[D(m(\mathcal{G}), m(\mathcal{G}_c)) \leq \epsilon\big] \geq 1 - \delta.$$

*where $D$ is a suitable distance operator between the outputs. For instance, it may quantify discrepancy in output performance, or the distance between output distributions, for example KL-divergence on the empirical logit distribution. We abstract these details for flexibility.*

Our proposed reformulation of the problem adheres to the following *necessary* conditions:

- **$\epsilon$-task preservation:** The distance constraint directly bounds the predictive performance degradation within $\epsilon$, ensuring reliable task retention after condensation.
- **Compactness:** The constraint $|\mathcal{G}_c| \leq B$ enforces that the byte-size of the condensed graph stays within a specified budget, encouraging practical usability. We advocate for bytes instead of node compression ratio since it holistically captures all aspects of node size, edge density and features.
- **Generalization:** The probabilistic formulation over $m \sim \mathcal{M}$ promotes robustness across model architectures, hyperparameters, and initializations, supporting broader applicability.

### 2.2. Desirable Properties

In addition to the core requirements, several *desirable* properties emerge from the broader goals of graph condensation:

- **Faithful representation.** The condensed graph should preserve the semantic meaning of nodes, edges, and features, ensuring that structural and attribute-based interpretations remain consistent with the original graph. (e.g. citations in a citation network)
- **Memory efficiency.** Condensation methods must scale to graphs with millions of nodes and edges without constructing dense $N^2$ intermediates, ensuring end-to-end memory usage remains within practical hardware limits.
- **Democratization and sustainability.** Algorithms should run efficiently on CPU-only infrastructure and leverage embarrassingly parallel computation on multi-core servers, reducing reliance on specialized GPUs and lowering environmental impact.

- **Fairness and robustness.** Condensation procedures should preserve fairness with respect to sensitive attributes and remain robust under adversarial or distributional shifts, for equitable and reliable inference across diverse scenarios.
- **Hyperparameter simplicity.** Condensation should be robust to hyperparameter choices, avoiding sensitivity to minor tuning and eliminating the need for exhaustive grid search.

## 3. Existing Works: Merits and Limitations

Table 1 organizes the landscape of a representative subset of graph condensation methods along two principal axes: (i) whether the condensation process is model-specific, and (ii) whether it requires training on the full dataset as a subroutine. These axes yield four distinct categories: (1) Model-dependent with full-dataset training, (2) Model-dependent without full-dataset training, (3) Model-agnostic with full-dataset training, (4) Model-agnostic without full-dataset training. Within each bucket, we further evaluate methods based on additional desiderata discussed in § 2.

Algorithms that require full-dataset training are costlier than training on the original graph, defeating condensation's scalability goal; aiding occasional NAS does not redeem them because NAS is computationally intensive and infrequently used, so it cannot justify full-dataset condensation.

Likewise, model-dependent approaches tightly couple the condensation objective to a specific architecture, thereby limiting applicability to tasks such as neural architecture search (NAS).

As such, Bucket (1) is the least desirable—it violates both key goals of graph condensation. Yet, it remains the dominant category in prior work, and therefore, serving as the primary motivation behind this position paper. In contrast, Bucket (4) satisfies both desiderata: model-agnosticism and independence from full-dataset training. It thus represents the most promising direction for practical, reusable graph condensation. Buckets (2) and (3), which relax only one of the constraints, offer partial progress and mark useful steps toward the ideal represented by Bucket (4).

We next discuss the source of these dependencies.

### 3.1. Model-Dependent Methods

Buckets (1) and (2) rely on signals from a specific model, producing condensed graphs that are inherently architecture-dependent. The most common source of this coupling is *gradient matching*. For instance, GCOND (Jin et al., 2022b) aligns gradient trajectories between the original and condensed graphs, leading to condensed graphs that often fail to generalize across architectures (Wang & Li,

*Table 1.* Comparison of methods across necessary and desirable conditions, partitioned by model-dependence and full-training requirement. The symbol ◯ denotes that the property has not been investigated; the green ✓ indicates support, and the red ✗ indicates non-support. Memory Effective Pipeline shows a red ✗ whenever intermediate $N'^2$ either Laplacian or adjacency matrices are materialized or when kernel methods are used due to $N^2$ kernel computation.

| Method | GPU/CPU | Grid Search | Agnostic | Faithful Rep. | Mem.-Eff. Pipeline | Fairness | Robustness |
|---|---|---|---|---|---|---|---|
| **Model-dependent & Full-training** | | | | | | | |
| GCOND (Jin et al., 2022b) | GPU | ✓ | ✗ | ✗ | ✗ | ◯ | ◯ |
| DOSCOND (Jin et al., 2022a) | GPU | ✓ | ✗ | ✗ | ✗ | ◯ | ◯ |
| MSGC (Gao & Wu, 2023) | GPU | ✓ | ✗ | ✓ | ✓ | ◯ | ◯ |
| FGD (Feng et al., 2023) | GPU | ✓ | ✗ | ✗ | ✗ | ✓ | ◯ |
| SFGC (Zheng et al., 2024) | GPU | ✓ | ✗ | ✗ | ✓ | ◯ | ◯ |
| GCARE (Mao et al., 2023) | GPU | ✓ | ✗ | ✗ | ✗ | ◯ | ◯ |
| GROC (Li et al., 2023) | GPU | ✓ | ✗ | ✗ | ✗ | ◯ | ◯ |
| FEDGKD (Pan et al., 2023) | GPU | ✓ | ✗ | ✗ | ✗ | ◯ | ◯ |
| GSTAM (Rasti-Meymandi et al., 2024) | GPU | ✗ | ✗ | ✗ | ✗ | ◯ | ◯ |
| EXGC (Fang et al., 2024) | GPU | ✓ | ✗ | ✗ | ✗ | ◯ | ◯ |
| GEOM (Zhang et al., 2024b) | GPU | ✓ | ✗ | ✗ | ✗ | ◯ | ◯ |
| MCOND (Gao et al., 2024a) | GPU | ✓ | ✗ | ✗ | ✗ | ◯ | ◯ |
| CTRL (Zhang et al., 2024a) | GPU | ✓ | ✗ | ✗ | ✗ | ◯ | ◯ |
| FEDGC (Zhang et al., 2025) | GPU | ✓ | ✗ | ✗ | ✗ | ◯ | ◯ |
| FEDGM (Yan et al., 2025) | GPU | ✓ | ✗ | ✗ | ✗ | ◯ | ◯ |
| SGSGC (Li et al., 2025a) | GPU | ✓ | ✗ | ✗ | ✗ | ◯ | ◯ |
| **Model-dependent & No Full-training** | | | | | | | |
| SGDC (Wang et al., 2024b) | GPU | ✓ | ✗ | ✗ | ✗ | ◯ | ◯ |
| KIDD (Xu et al., 2023) | GPU | ✓ | ✗ | ✗ | ✗ | ◯ | ◯ |
| GC-SNTK (Wang et al., 2024a) | GPU | ✓ | ✗ | ✗ | ✗ | ◯ | ◯ |
| CAT (Liu et al., 2023) | GPU | ✓ | ✗ | ✗ | ✓ | ◯ | ◯ |
| PUMA (Liu et al., 2024b) | GPU | ✓ | ✗ | ✗ | ✓ | ◯ | ◯ |
| ST-GCOND (Yang et al., 2025) | GPU | ✓ | ✗ | ✗ | ✗ | ◯ | ◯ |
| **Model-agnostic & Full-training** | | | | | | | |
| GDEM (Liu et al., 2024a) | GPU | ✓ | ✓ | ✗ | ✗ | ◯ | ◯ |
| BIMSGC (Fu et al., 2025) | GPU | ✓ | ✓ | ✗ | ✗ | ◯ | ◯ |
| CTGC (Gao et al., 2025a) | GPU | ✓ | ✓ | ✗ | ✗ | ◯ | ◯ |
| CLUSTGDD (Lai et al., 2025) | GPU | ✗ | ✓ | ✗ | ✗ | ◯ | ◯ |
| **Model-agnostic & No Full-training** | | | | | | | |
| GCDM (Liu et al., 2022a) | GPU | ✓ | ✓ | ✗ | ✗ | ◯ | ◯ |
| DISCO (Xiao et al., 2025) | GPU | ✓ | ✓ | ✓ | ✓ | ◯ | ◯ |
| OPENGC (Gao et al., 2024b) | GPU | ✓ | ✓ | ✗ | ✗ | ◯ | ◯ |
| GCPA (Li et al., 2025b) | GPU | ✗ | ✓ | ✓ | ✓ | ◯ | ◯ |
| TMD (Jain et al., 2024) | CPU | ✗ | ✓ | ✓ | ✓ | ◯ | ◯ |
| MIRAGE (Gupta et al., 2024) | CPU | ✗ | ✓ | ✓ | ✓ | ◯ | ◯ |
| BONSAI (Gupta et al., 2025b) | CPU | ✗ | ✓ | ✓ | ✓ | ◯ | ◯ |

2025; Gupta et al., 2025b). Bucket (1) methods are particularly costly—condensation often takes longer than training a model on the original graph—and are highly sensitive to hyperparameters.

Several follow-up methods inherit these limitations, including DOSCOND (Jin et al., 2022a), MSGC (Gao & Wu, 2023), EXGC (Fang et al., 2024), CTRL (Zhang et al., 2024a), MCOND (Gao et al., 2024a), and GCSR (Liu et al., 2024c). Methods such as SFGC (Zheng et al., 2024) and GEOM (Zhang et al., 2024b) attempt to improve fidelity by aligning thousands of gradient trajectories, but this drastically increases computational overhead and deepens the model-specific entanglement.

Some approaches (GCSR, GEOM, MCOND) target interpretability, lossless condensation, and inductive capability; while gradients can encode task-level signals that sometimes transfer, they often mix in architecture-/optimizer-/initialization-specific components, so unless a method is explicitly gradient-agnostic or filters model-specific leakage, it remains costly and brittle.

A few model-dependent methods use trained or initialized GNNs indirectly. For example, GSTAM (Rasti-Meymandi et al., 2024) aligns attention maps between the original and synthetic graphs. However, this approach is still shaped by the model's inductive biases, limiting its ability to generalize across architectures.

Bucket (2) methods improve compute efficiency but remain architecturally dependent. Kernel-based approaches such as KIDD, GC-SNTK, and SGDC rely on architecture-specific kernels, limiting portability, hindering uses such as neural architecture search, and incurring high memory costs (Gao et al., 2025b). ST-GCOND shares these limitations but is motivated by a MAML-style multi-task formulation that amortizes higher upfront cost by producing condensed datasets intended for reuse across tasks.

In summary, Bucket (1) methods suffer from high computational cost and poor generalization, undermining the primary goals of graph condensation. Bucket (2) methods are more efficient but remain constrained by their model-specific formulations, limiting broader applicability.

### 3.2. Model-Agnostic Methods

Buckets (3) and (4) decouple the condensation process from any specific architecture, thereby improving generalization. A widely adopted strategy for achieving this independence is *distribution matching*—rather than gradient matching—over the space of computation trees (Gupta et al., 2024; 2025b; Jain et al., 2024), leveraging the observation that GNN embeddings primarily depend on $L$-hop computation trees.

TMD (Jain et al., 2024) minimizes Tree Mover's Distance (Chuang & Jegelka, 2022) between computation-tree distributions of the original and subsampled graphs. The method is efficient, interpretable, and model-agnostic, with strong performance and generalization, but evaluations are limited to graph classification and report compression by node count rather than byte size.

MIRAGE (Gupta et al., 2024) identifies frequent computation trees and condenses the graph by retaining high-frequency patterns. It is fast, interpretable, and model-agnostic, producing sparse condensed graphs with strong performance. Nonetheless, its reliance on exact tree isomorphisms limits applicability to graphs with continuous node features or high-degree nodes. The method also primarily targets binary graph classification tasks.

BONSAI (Gupta et al., 2025b) builds on MIRAGE, selecting exemplar computation trees that are both representative—via reverse-$k$NN in Weisfeiler-Lehman (WL) embedding space—and diverse—via greedy max-coverage selection. This yields a small, structurally informative subset that captures the graph's essential variation. Both MIRAGE and BONSAI report compression in terms of byte size, providing a more accurate measure of information density than node count.

GCPA further simplifies this direction by using lightweight neighborhood aggregation similar to BONSAI, achieving 96×–2455× speedups over GDEM; however, because it does not synthesize edges, it fails for attention-based models such as GAT, where edge attention weights are never trained.

While methods like BONSAI and MIRAGE avoid gradient dependence, they still make implicit architectural assumptions by relying on $L$-hop computation trees that require specifying the number of layers $L$. Although this represents a softer constraint than explicit gradient matching—since distant neighbor information diminishes due to over-squashing—truly model-agnostic condensation should eliminate architectural dependencies entirely.

GCDM (Liu et al., 2022a) also operates in the space of receptive fields akin to computation trees, matching their distributions using Maximum Mean Discrepancy (MMD). Similarly, CAT (Liu et al., 2023) and PUMA (Liu et al., 2024b) employ MMD with randomly initialized GNNs in continual learning settings. DISCO (Xiao et al., 2025) takes a modular approach, separately condensing nodes and edges by learning node mappings and link predictors to preserve the semantic structure of the original graph.

Bucket (3) methods are model-agnostic and avoid full-graph GNN training. Approaches such as GDEM, BIMSGC, and CTGC leverage eigenbasis matching to align graph structure, coupled with lightweight MLP-based optimization. CLUSTGDD adopts a different strategy, learning node embeddings with an MLP, clustering them to generate synthetic features, and constructing edges via embedding-similarity reweighting and sampling. Collectively, these methods illustrate a promising middle ground between structural fidelity and practical efficiency.

Methods such as DISCO, TMD, MIRAGE, and BONSAI preserve the semantic and distributional structure of the original graph by selecting representative nodes and edges, often via computation-tree proxies. This yields more faithful condensed graphs than model-dependent approaches. However, reliance on $l$-hop computation trees in BONSAI and MIRAGE introduces a soft dependence on GNN depth that ideally should be avoided.

Among existing approaches, **ST-GCOND is the only method that is explicitly *task-agnostic***, aiming to amortize condensation cost across multiple downstream tasks. This direction remains underexplored. Future work should prioritize extending efficient, model-agnostic methods such as GCPA, TMD, BONSAI, and GCDM toward similar task-agnostic formulations, enabling broader reuse without reintroducing model- or task-specific coupling.

## 4. Call for Action

**Benchmark What Actually Needs Condensation:** A critical flaw in current graph condensation research lies in the near-universal reliance on small, outdated datasets—such as Cora, CiteSeer, and PubMed—that are computationally

*Table 2.* Scalability on MAG240M: gradient-based condensation methods cannot scale to large graphs (OOM/OOT).

| $S_r$ (%) | Random | Herding | GCOND | GDEM | GCSR | ExGC | GC-SNTK | GEOM | BONSAI |
|---|---|---|---|---|---|---|---|---|---|
| 0.5 | 34.43±0.11 | 36.18±0.10 | OOT | OOM | OOT | OOT | OOT | OOT | 52.33±0.05 |
| 1.0 | 37.93±0.08 | 37.29±0.06 | OOT | OOM | OOT | OOT | OOT | OOT | 52.58±0.33 |
| 3.0 | 42.96±0.43 | 37.98±0.09 | OOT | OOM | OOT | OOT | OOT | OOT | 53.39±0.07 |

trivial. These graphs, with a few thousand nodes, can be trained on in seconds with modern hardware. Worse, as shown by (Huang et al., 2022), even randomly initialized GNNs can outperform trained models on such datasets, making condensation research on them effectively redundant. Further, (Katsman et al., 2024) show that in many of these cases, graph structure contributes little beyond raw features, casting doubt on the value of any structure-preserving condensation in these settings.

Datasets that genuinely require condensation due to scale remain underexplored. MAG240M (Hu et al., 2021), with hundreds of millions of nodes and edges, exemplifies the regime where full-graph training is infeasible and condensation is necessary. Yet at this scale, limitations are clear: Table 2 shows only BONSAI succeeds, while others fail due to reliance on full-graph training (Jin et al., 2022b; Liu et al., 2024c; Zhang et al., 2024b; Fang et al., 2024; Wang et al., 2024a) or costly operations such as Laplacian construction (Liu et al., 2024a). Table 7 further shows gradient-based methods incur substantially higher $CO_2$ emissions, up to 12.4× higher for GCOND on Reddit. This mismatch between method design and target scale has led to misleading conclusions about scalability.

We argue that future research must anchor itself in use cases where condensation is not optional but necessary. This entails:

- Focusing evaluation on large-scale benchmarks (e.g., MAG240M (Hu et al., 2021), MalNet (Freitas et al., 2021), TUDataset (Morris et al., 2020), force fields (Chmiela et al., 2017), PDNS-Net (Kumarasinghe et al., 2022), OMAT24 (Barroso-Luque et al., 2024), MPTrj (Deng et al., 2023), Alexandria (Schmidt et al., 2023)) where full-graph training is prohibitively expensive.
- Incorporating synthetic datasets with tunable size and complexity to study controlled trade-offs between graph size, structure preservation, and downstream performance.

This reorientation would discourage full-graph training methods that are infeasible where condensation is needed. Evaluating realistic workloads can shift the field from proof-of-concept prototypes toward scalable, deployable methods.

**Towards Honest and Transparent Metrics:** The dominant practice of measuring graph condensation effectiveness by node count reduction masks the true computational and memory demands of the resulting graph, and creates opportunities for inflated claims of efficiency. This problem is starkly illustrated when examining actual storage requirements. We constructed 1% condensed graphs of ogbn-arxiv with various methods, where 1% represents whatever each method defines as 1%—node ratio for most, byte size ratio for BONSAI. Disk sizes in Table 3 reveal major inconsistencies: storage ranges from 644KB to 3.66MB, a 5.7× difference despite identical "1% compression" claims. This byte-size variability exposes fundamentally different information capacities, invalidating direct performance comparisons between methods.

*Table 3.* Condensed graph sizes (1% condensation) on `ogbn-arxiv`.

| Method | GDEM | BONSAI | GCOND | GCSR |
|---|---|---|---|---|
| Size | 3.66MB | 1.20MB | 3.65MB | 3.66MB |
| Method | ExGC | GC-SNTK | GEOM | Full dataset |
| Size | 2MB | 644KB | 2.99MB | ≈87MB |

More concerning is the widespread practice of reporting condensation ratios relative to the full dataset when only the training split is used in condensation (Jin et al., 2022b; Gao et al., 2024a; Feng et al., 2023; Liu et al., 2024c; Li et al., 2023; Wang et al., 2024a). Consider Cora: although it has 2,708 nodes, typical training uses just 140. If a method condenses those 140 to 139 nodes and reports a "5% condensation," it implies reducing the entire graph to 5%, when in fact virtually no reduction occurred. This obfuscates two issues: (1) it vastly overstates the level of achieved compression, and (2) it hides the gap in performance between train-time and full-graph evaluation, especially when generalization matters.

Equally problematic is the neglect of condensation time itself. Many state-of-the-art methods involve prolonged optimization phases that far exceed the time required to train on the full dataset. Yet, this overhead is often omitted in reported metrics, giving an illusion of efficiency (Jin et al., 2022b;a; Liu et al., 2024c; Zheng et al., 2024; Yang et al., 2023a). This problem becomes starkly evident in Table 6; gradient-based techniques that embed full-graph training as a subroutine frequently surpass the running time of end-to-end GNN training and, in some cases, fail to complete within standard time limits.

To move beyond such superficial metrics, we advocate for a shift toward more transparent and holistic evaluation practices:

- **Byte-level compression ratio:** Report the total memory footprint of the condensed graph, including node features, edge indices, and auxiliary data. This directly reflects real-world storage and deployment constraints.
- **Disaggregated compression statistics:** Separately report node ratio, edge ratio, and average feature sparsity, each relative to the actual training graph, not the full dataset.
- **Total computational cost:** Always report both the condensation time and the training time on the condensed graph. Omitting the often expensive condensation phase distorts the practical value of the method—especially when condensation takes longer than training the original model.
- **Hyperparameter Sensitivity and Tuning Overhead:** Many graph condensation algorithms rely on a large number of hyperparameters. While having tunable parameters is not inherently a drawback, the practical challenge arises when their performance is highly sensitive to these settings and tuning requires exhaustive grid search. This significantly inflates the total computational cost and poses a serious barrier to real-world deployment, where extensive hyperparameter tuning is often impractical. To address this, future work should systematically evaluate the sensitivity of condensation performance to hyperparameter choices, offer principled heuristics or adaptive schemes for setting them, and favor methods that remain robust across a range of settings with minimal tuning effort.

**Move Beyond Gradient-Based Methods for Architecture-Agnostic Condensation:** The architectural brittleness of gradient-based condensation methods demands a pivot away from this paradigm. Table 4 shows that gradient-matching methods fail catastrophically under architecture transfer: GCOND drops from 77.30% to 13.21% when moving from GCN to GAT on Cora, and GCSR collapses from 79.94% to 32.01% on PubMed. These are not marginal degradations but system failures that preclude practical deployment where architectural flexibility is required. GC-Bench (Sun et al., 2024) and GC4NC (Gong et al., 2024) similarly report widespread transfer failures, particularly to attention-based models such as GAT, and further note that NAS benefits from condensation only when coupled with expensive inner-loop or long-horizon trajectory optimization. This brittleness is expected given the disparate inductive biases of GNNs—mean aggregation in GCN, sum pooling in GIN, and attention in GAT. Nevertheless, some works obscure this limitation by restricting evaluation to closely related architectures (e.g., SGC and GCN) (Jin et al., 2022b;a; Fang et al., 2024). Since recent methods such as GCPA and ST-GCOND postdate these benchmarks, updated evaluations are needed to reassess architecture-agnostic condensation.

**Invest in Theoretical Foundations:** Despite rapid empirical progress, the theoretical underpinnings of graph condensation remain weakly explored. Existing algorithms often ignore the computational complexity of their proposed condensation scheme, making it unclear how different condensation strategies compare to full-graph training in terms of time, space, and resource overhead. For model-dependent methods, particularly those justified via Neural Architecture Search (NAS), there is little theoretical grounding to support the idea that gradients or trajectories from one architecture can effectively guide search over others.

Moreover, the formal guarantees for condensation remain elusive. As posed in our formulation (§ 2), can we design condensation algorithms that minimize the worst-case performance gap $\epsilon$ with high probability $(1 - \delta)$ under strict byte-budget constraints $B$? Establishing such guarantees over model classes $\mathcal{M}$, possibly under realistic assumptions (e.g., smoothness of training dynamics or stability of learned representations), would provide much-needed clarity on the achievable trade-offs between graph size, generalization, and fidelity. The field would benefit from principled lower bounds, approximation guarantees, and a formal complexity landscape for this increasingly influential task.

## 5. Alternative Views

To ensure a balanced discussion, we briefly acknowledge arguments that defend existing condensation practices:

**Architecture-Specific Efficiency** In many production pipelines, a single GNN architecture is standardized, making architecture-specific condensates more valuable than general-purpose ones. For stable, well-defined tasks—such as recommendation or fraud detection—marginal performance gains across diverse architectures may not justify underperformance on the deployed model. For example, a graph condensed via full training on *SGC* (Wu et al., 2019) might perform well only on *GCN* (Kipf & Welling, 2017) but fail on *GAT*; this may still be acceptable if *GCN* does not scale to the original graph while *SGC* does.

**One-Time Computation for Broad Reuse** Large organizations may amortize the high one-time cost of condensation by distributing the resulting synthetic graph internally. These fixed condensates can accelerate downstream experimentation—such as hyperparameter sweeps—without repeatedly training on the full graph. Even if the condensed graph generalizes poorly across architectures, it may still serve as a lightweight proxy: one could randomize hyperparameters, condense using a small model, and then perform fast tuning before full-scale training. While model-agnostic approaches offer similar benefits, this helps explain why slow or architecture-specific methods may still have practical utility.

*Table 4.* Accuracies (%) of various methods on GCN, GAT, and GIN across label rates. A trend appears showing the fall of accuracy when going from GCN to GAT or GIN for model-dependent methods. This is because GCN was the original model used to construct the synthetic datasets. The performance trade-off is much better for model-agnostic methods. We display a fall of $< 10\%$ as↓and a fall of larger than 10% as↓↓while↑indicates improvements. Our time-out (OOT) threshold is 5 hours, memory out-of-memory (OOM) is 40GB. See Table 5 for results on more datasets.

| Dataset | % | GNN | Random | Herding | GCond | GDEM | GCSR | Bonsai | Full |
|---|---|---|---|---|---|---|---|---|---|
| Flickr | 0.5 | GCN | 44.78±0.00 | 47.98±0.01 | 44.06±1.05 | 46.25±1.02 | 46.41±0.00 | 48.73±0.27 | 50.93±0.17 |
| | 1.0 | GCN | 44.21±0.03 | 46.72±0.01 | 39.88±5.60 | 46.99±1.38 | OOT | 49.05±0.17 | 50.93±0.17 |
| | 3.0 | GCN | 46.56±0.01 | 46.54±0.01 | 46.04±1.88 | 47.35±0.98 | OOT | 49.66±0.27 | 50.93±0.17 |
| | 0.5 | GAT | 43.64±0.99 | 36.50±13.22 | 40.24±3.20↓ | 25.43±10.37↓↓ | 28.03±6.60↓↓ | 48.22±3.60↓ | 51.42±0.07 |
| | 1.0 | GAT | 43.56±1.06 | 36.34±1.14 | 40.85±1.08↑ | 18.44±9.42↓↓ | OOT | 45.62±1.85↓ | 51.42±0.07 |
| | 3.0 | GAT | 45.71±1.87 | 42.70±1.17 | 41.51±9.81↓ | 25.83±11.39↓↓ | OOT | 47.80±2.06↓ | 51.42±0.07 |
| | 0.5 | GIN | 42.67±0.83 | 39.98±7.21 | 13.65±7.54↓↓ | 14.10±5.68↓↓ | 05.92±1.01↓↓ | 44.97±2.23↓ | 45.37±0.57 |
| | 1.0 | GIN | 42.90±0.76 | 41.87±4.52 | 16.65±6.55↓↓ | 19.44±9.68↓↓ | OOT | 44.90±0.88↓ | 45.37±0.57 |
| | 3.0 | GIN | 19.63±4.21 | 43.72±3.26 | 24.25±14.43↓↓ | 20.97±6.64↓↓ | OOT | 45.04±1.94↓ | 45.37±0.57 |

**Value of Rich Training Signals** Matching gradient trajectories captures fine-grained loss landscape information that simple distribution- or tree-matching may miss. Existing benchmarks often avoid harder condensation tasks (e.g., (Yang et al., 2016)). For complex datasets like ZINC, PATTERN, and CLUSTER from (Dwivedi et al., 2023), which are not large but pose significant learning challenges, repeated passes of noise removal—achievable via gradient-based methods—could be advantageous. This aligns with evidence from (Li et al., 2020) showing that deep models initially learn correct patterns while ignoring noisy labels.

## 6. Conclusion

Graph condensation risks becoming an elegant solution in search of a problem. Although promoted to accelerate training and reduce memory, many existing methods rely on full-graph training, gradient matching, and architecture-specific tuning, making them slower, less scalable, and harder to deploy than the models they aim to simplify. Gradient-based condensation is best viewed as architecture- and inductive-bias–specific dataset distillation. It can accelerate training within a narrow model family but is ill-suited for cross-architecture transfer, particularly to attention-based GNNs such as GAT. In contrast, non-gradient methods are more robust and scalable, making them preferable in large-scale or heterogeneous settings. This reflects a broader issue: methods are evaluated on benchmarks where condensation is unnecessary, and where efficiency metrics misrepresent real costs, skewing progress toward algorithmic sophistication over utility. To move beyond academic curiosity, the field must confront settings where full-graph training is infeasible, redefine compression in terms of real resource savings, and design methods that are robust, architecture-agnostic, and fast to compute. This requires rethinking what counts as success in graph condensation.

## Impact Statement

This position paper critically examines current graph condensation practices and advocates for more efficient, model-agnostic, and deployable approaches. Its primary impact is methodological: clarifying pitfalls, discouraging misleading evaluation practices, and guiding future research toward more realistic and resource-aware benchmarks. We do not foresee direct societal or ethical consequences beyond those already associated with research on graph neural networks.

## Acknowledgements

We acknowledge the Yardi School of AI, IIT Delhi for supporting this research. Sayan Ranu acknowledges the Nick McKeown Chair position endowment. Hariprasad Kodamana acknowledges IIT Delhi Abu Dhabi for partial support of this research. Mridul Gupta acknowledges the generous grant received from Yardi School of AI, IIT Delhi to sponsor his travel to ICML 2026.

### Code Availability

We release the scripts and integration code used to generate the experiments and analyses in this paper, together with links to the original implementations on which our pipeline builds at the following URL: https://github.com/idea-iitd/Graph-Condensation-Position-paper

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

# A. Technical Appendices and Supplementary Material

All experiments were conducted on a Linux-based server equipped with a 40GB NVIDIA A100 GPU, 96 Intel Xeon Gold 6248 CPU cores, and 512GB of RAM. For each method evaluated, the original codebases released by the respective authors were used.

*Table 5.* Accuracies (%) of various methods on GCN, GAT, and GIN across label rates. A trend appears showing the fall of accuracy when going from GCN to GAT or GIN for model-dependent methods. This is because GCN was the original model used to construct the synthetic datasets. The performance trade-off is much better for model-agnostic methods. We display a fall of $< 10\%$ as↓and a fall of larger than 10% as↓↓while↑indicates improvements.

| Dataset | % | GNN | Random | Herding | GCond | GDEM | GCSR | Bonsai | Full |
|---|---|---|---|---|---|---|---|---|---|
| Cora | 0.5 | GCN | 39.90±1.46 | 45.61±0.01 | 77.30±0.30 | 57.49±6.87 | 74.83±0.75 | 83.95±0.39 | 88.56±0.18 |
| | 1.0 | GCN | 27.75±2.05 | 52.07±0.00 | 77.30±0.31 | 69.32±4.71 | 77.56±0.74 | 85.76±0.24 | 88.56±0.18 |
| | 3.0 | GCN | 52.26±1.69 | 67.60±0.00 | 81.73±0.48 | 81.70±3.10 | 77.20±0.48 | 86.38±0.22 | 88.56±0.18 |
| | 0.5 | GAT | 41.44±1.73 | 33.80±0.07 | 13.21±1.99↓↓ | 63.91±5.91↑ | 15.09±6.19↓↓ | 75.42±1.61↓ | 85.70±0.09 |
| | 1.0 | GAT | 42.73±1.03 | 46.09±0.86 | 35.24±0.00↓↓ | 73.49±2.64↑ | 37.60±1.34↓↓ | 78.67±0.89↓ | 85.70±0.09 |
| | 3.0 | GAT | 60.22±0.67 | 56.75±0.45 | 35.24±0.00↓↓ | 75.28±4.86↓ | 36.72±0.81↓↓ | 80.66±0.80↓ | 85.70±0.09 |
| | 0.5 | GIN | 49.04±0.50 | 34.39±1.03 | 14.13±6.80↓↓ | 63.65±7.11↑ | 76.05±0.44↑ | 85.42±0.74↑ | 86.62±0.28 |
| | 1.0 | GIN | 50.48±0.85 | 33.80±2.42 | 33.91±1.23↓↓ | 75.92±4.24↑ | 60.70±4.44↓↓ | 84.80±0.41↓ | 86.62±0.28 |
| | 3.0 | GIN | 59.52±0.88 | 36.35±0.59 | 31.70±4.97↓↓ | 59.59±7.95↓↓ | 51.62±5.00↓↓ | 85.42±0.53↓ | 86.62±0.28 |
| Citeseer | 0.5 | GCN | 33.90±2.16 | 22.82±0.00 | 74.17±0.68 | 70.05±2.40 | 67.03±0.61 | 77.00±0.15 | 78.53±0.15 |
| | 1.0 | GCN | 44.90±2.32 | 49.10±0.02 | 77.62±0.71 | 72.48±2.13 | 74.77±0.78 | 77.03±0.33 | 78.53±0.15 |
| | 3.0 | GCN | 44.50±1.27 | 67.69±0.01 | 77.02±0.22 | 76.20±0.55 | 77.27±0.28 | 75.89±0.26 | 78.53±0.15 |
| | 0.5 | GAT | 42.76±0.35 | 36.04±0.46 | 21.47±0.00↓↓ | 69.86±2.28↓ | 21.92±0.76↓ | 68.56±0.57↓ | 77.48±0.75 |
| | 1.0 | GAT | 46.19±1.38 | 52.07±0.11 | 21.47±0.00↓↓ | 23.87±3.05↓↓ | 21.50±0.06↓↓ | 69.43±0.82↓ | 77.48±0.75 |
| | 3.0 | GAT | 61.65±0.51 | 65.17±0.00 | 21.26±0.22↓↓ | 22.90±1.20↓↓ | 21.50±0.06↓↓ | 69.94±1.15↓ | 77.48±0.75 |
| | 0.5 | GIN | 44.86±0.43 | 22.97±0.30 | 21.47±0.00↓↓ | 67.69±3.28↓ | 50.66±1.17↓↓ | 71.80±0.26↓ | 75.45±0.23 |
| | 1.0 | GIN | 47.90±0.65 | 39.67±0.82 | 19.49±1.09↓↓ | 67.64±4.45↓ | 64.74±1.88↓↓ | 72.16±0.60↓ | 75.45±0.23 |
| | 3.0 | GIN | 61.83±0.68 | 60.48±0.26 | 18.65±2.56↓↓ | 48.65±8.17↓↓ | 59.95±9.07↓↓ | 70.51±0.54↓ | 75.45±0.23 |
| PubMed | 0.5 | GCN | 62.58±0.25 | 78.29±0.00 | 80.63±1.20 | 80.72±0.92 | 79.43±0.25 | 87.27±0.03 | 87.22±0.00 |
| | 1.0 | GCN | 79.19±0.09 | 78.59±0.00 | 79.92±0.00 | 80.80±1.07 | 79.11±0.15 | 87.08±0.04 | 87.22±0.00 |
| | 3.0 | GCN | 82.50±0.09 | 78.09±0.00 | 77.00±0.15 | 81.07±0.90 | 79.94±0.16 | 87.64±0.09 | 87.22±0.00 |
| | 0.5 | GAT | 77.73±0.12 | 75.44±0.02 | 37.49±4.01↓↓ | 80.06±1.16↓ | 38.29±8.13↓↓ | 85.66±0.38 | 86.33±0.08 |
| | 1.0 | GAT | 78.85±0.09 | 76.64±0.02 | 41.55±3.18↓↓ | 80.75±0.47↓ | 40.47±0.00↓↓ | 85.88±0.28↓ | 86.33±0.08 |
| | 3.0 | GAT | 82.84±0.11 | 78.48±0.03 | 37.77±3.61↓↓ | 65.08±9.53↓ | 40.27±0.20↓↓ | 85.62±0.36↓ | 86.33±0.08 |
| | 0.5 | GIN | 77.45±0.14 | 48.48±1.33 | 30.91±4.57↓↓ | 78.78±0.91↓ | 36.88±12.06↓↓ | 84.32±0.33↓ | 84.66±0.05 |
| | 1.0 | GIN | 78.43±0.22 | 62.22±0.13 | 32.84±6.27↓↓ | 78.72±0.95↓ | 33.75±5.58↓↓ | 85.57±0.26↓ | 84.66±0.05 |
| | 3.0 | GIN | 80.56±0.17 | 45.40±0.46 | 36.11±3.47↓↓ | 81.08±0.99↑ | 32.01±6.77↓↓ | 85.66±0.23↓ | 84.66±0.05 |
| Flickr | 0.5 | GCN | 44.78±0.00 | 47.98±0.01 | 44.06±1.05 | 46.25±1.02 | 46.41±0.00 | 48.73±0.27 | 50.93±0.17 |
| | 1.0 | GCN | 44.21±0.03 | 46.72±0.01 | 39.88±5.60 | 46.99±1.38 | OOT | 49.05±0.17 | 50.93±0.17 |
| | 3.0 | GCN | 46.56±0.01 | 46.54±0.01 | 46.04±1.88 | 47.35±0.98 | OOT | 49.66±0.27 | 50.93±0.17 |
| | 0.5 | GAT | 43.64±0.99 | 36.50±13.22 | 40.24±3.20↓ | 25.43±10.37↓↓ | 28.03±6.60↓↓ | 48.22±3.60↓ | 51.42±0.07 |
| | 1.0 | GAT | 43.56±1.06 | 36.34±1.14 | 40.85±1.08↑ | 18.44±9.42↓↓ | OOT | 45.62±1.85↓ | 51.42±0.07 |
| | 3.0 | GAT | 45.71±1.87 | 42.70±1.17 | 41.51±9.81↓ | 25.83±11.39↓↓ | OOT | 47.80±2.06↓ | 51.42±0.07 |
| | 0.5 | GIN | 42.67±0.83 | 39.98±7.21 | 13.65±7.54↓↓ | 14.10±5.68↓↓ | 05.92±1.01↓↓ | 44.97±2.23↓ | 45.37±0.57 |
| | 1.0 | GIN | 42.90±0.76 | 41.87±4.52 | 16.65±6.55↓↓ | 19.44±9.68↓↓ | OOT | 44.90±0.88↓ | 45.37±0.57 |
| | 3.0 | GIN | 19.63±4.21 | 43.72±3.26 | 24.25±14.43↓↓ | 20.97±6.64↓↓ | OOT | 45.04±1.94↓ | 45.37±0.57 |

## A.1. Carbon Emissions

*Table 6.* Condensation runtimes (in seconds) vs. full-graph training ("Full"). "OOT" = out-of-time/-of-memory.

| Dataset | GCond | GDEM | GCSR | ExGC | GC-SNTK | GEOM | BONSAI | Full |
|---|---|---|---|---|---|---|---|---|
| Cora | 2 738 | 105 | 5 260 | 34.87 | 82 | 12 996 | 2.60 | 24.97 |
| Citeseer | 2 712 | 167 | 6 636 | 34.51 | 124 | 15 763 | 2.75 | 24.87 |
| Pubmed | 2 567 | 530 | 1 319 | 114.96 | 117 | OOT | 24.24 | 51.06 |
| Flickr | 1 935 | 3 405 | 17 445 | 243.28 | 612 | OOT | 118.23 | 180.08 |
| Ogbn-arxiv | 14 474 | 569 | OOT | 1 594.83 | 12 218 | OOT | 298.64 | 524.67 |
| Reddit | 30 112 | 20 098 | OOT | 6 903.47 | 29 211 | OOT | 1 170.64 | 2 425.68 |

*Table 7.* Estimated $CO_2$ emissions from condensation of various methods in seconds at 0.5%. $CO_2$ emissions are computed as 10.8kg per 100 hours for Nvidia A100 GPU and 4.32kg per 100 hours for 10 CPUs of Intel Xeon Gold 6248 (Lacoste et al., 2019).

| Dataset | GCond | GDEM | GCSR | ExGC | GC-SNTK | GEOM | BONSAI | Full |
|---|---|---|---|---|---|---|---|---|
| Cora | 82.14 | 3.15 | 157.80 | 1.05 | 2.46 | 389.88 | 0.03 | 0.75 |
| Citeseer | 81.36 | 5.01 | 199.08 | 1.03 | 3.72 | 472.89 | 0.03 | 0.75 |
| Pubmed | 77.01 | 15.90 | 39.57 | 3.45 | 3.51 | $\geq$540.00 | 0.3 | 1.53 |
| Flickr | 58.05 | 102.15 | 523.35 | 7.30 | 18.36 | $\geq$540.00 | 1.42 | 5.40 |
| Ogbn-arxiv | 434.22 | 17.07 | $\geq$540.00 | 46.49 | 366.54 | $\geq$540.00 | 4.18 | 15.74 |
| Reddit | 903.36 | 602.94 | $\geq$540.00 | 207.10 | 876.33 | $\geq$540.00 | 17.34 | 72.77 |

