# OpenReview forum: "Position: Graph Condensation Needs a Reset—Move Beyond Full-dataset Training and Model-Dependence"
_ICML.cc/2026/Position_Paper_Track — ICML 2026 Position Paper Track spotlight_

### Official Review · Reviewer_AmUP · 2026-02-18

**Significance:** 2
**Argument Clarity:** 3
**Rating:** 4
**Confidence:** 4

**Questions:**

1. In definition 2.2, the authors propose a reformulation of the graph condensation. The new formulation is stricter if I understand correctly. What is the benefit of doing so?

2. Do smaller byte-sizes guarantee faster training time?

**Alternative Views Section:**

Yes

**Compliance With Llm Reviewing Policy A Conservative:**

Affirmed.

**Discussion Potential:**

2

**Final Justification:**

I have increased the score. For the revision, I suggest adding a more detailed discussion on the practical use of dataset condensation in the vision community, and whether it could inspire real-world applications of graph condensation. In contrast, it is less important to emphasize how graph condensation introduces novel technical challenges absent in vision, or which techniques are inherited from the vision domain.

**Paper Summary:**

This position paper points out some issues in the research of graph condensation. For instance, many graph condensation methods require full-graph training, which goes against the purpose of graph condensation. There are other issues, like the codensation ratio is not necessarily the byte-wise shrinkage ratio. The authors also call for better benchmarks and transparent metrics.

**Position:**

Yes

**Position In Title:**

Yes

**Related Work:**

2

**Strengths And Weaknesses:**

Strengths:

1. I can confirm the claim of this paper, most of them match my experience.

2. The paper covers most of the recent works in graph condensation, the authors seem to be very familiar with this field.

Weaknesses:

1. The audience of graph condensation is not broad, so unfortunately, it is not very likely to inspire discussion. Graph condensation is about the efficiency of graph neural networks, but when their effectiveness is questionable, fewer people will focus on the training time/memory.

2. Graph condensation is built entirely on data condensation, which is a research direction mainly in the vision community. This position paper would benefit from more discussion on the situation of data condensation.

3. The suggested actions in the call to action section are not very insightful. For example, the first call is to use larger benchmarks, but I believe the community is aware that large graph condensation is more important, the problem is how to scale these methods.

**Support:**

4

---

> ### Author Rebuttal · Authors · 2026-03-27
>
> We thank the reviewer for the thoughtful engagement with our work. We address
> each concern below and hope our responses demonstrate the strength and
> significance of our contributions.
>
> ---
>
> > **W1. The audience of graph condensation is not broad...**
>
> We respectfully disagree and present the evidence behind our conviction below.
>
> GNNs are broad enough to feature as an explicit topic in the ICML call for papers. GNNs remain the dominant paradigm for learning on relational data including drug discovery [1], fraud detection [2], recommender systems [3], and knowledge graph reasoning [4]. The scalability of GNN training on real-world graphs in these domains is an active and pressing problem. MAG240M, with hundreds of millions of nodes and billions of edges, is representative of graphs that practitioners encounter routinely. As outlined in Table 1, 33 papers have been published within the last 3.5 years in premier ML venues, accumulating thousands of citations.
>
> [1] Zhang et al. (2025). Graph Neural Networks in Modern AI-Aided Drug Discovery. Chemical Reviews.
>
> [2] Motie et al. (2024). Financial Fraud Detection Using Graph Neural Networks: A Systematic Review. Expert Systems with Applications.
>
> [3] Wu et al. (2022). Graph Neural Networks in Recommender Systems: A Survey. ACM Computing Surveys.
> [4] Galkin et al. Towards Foundation Models for Knowledge Graph Reasoning. ICLR 2024.
>
> > **W2. This position paper would benefit from more discussion on the situation of data condensation.**
>
> Graph condensation shares foundational ideas with data distillation in the vision community — gradient matching in GCOND draws from Zhao et al. (2021), and trajectory matching in GEOM and SFGC adapts Cazenavette et al. (2022). Yet graph condensation has diverged distinctively, with techniques in Sec. 3.2 leveraging graph-specific ideas: compression of computation trees (BONSAI, TMD, MIRAGE, GCPA, GCDM), eigenbasis matching of the graph Laplacian (GDEM, CTGC, BiMSGC), and random weight initialization in GNNs (CAT, PUMA). We commit to expanding the related work section to cover both the inherited techniques and these graph-specific departures.
>
> > **W3. The suggested actions in the call to action section are not very insightful. For example, the first call is to use larger benchmarks, but I believe the community is aware that large graph condensation is more important, the problem is how to scale these methods.**
>
> We distinguish between the community being aware of a problem and acting on it. If awareness were translating into action, we would not observe the near-universal reliance on Cora, CiteSeer, and PubMed in published condensation benchmarks — graphs trainable in seconds, and the absence of MAG240m.
>
> Furthermore, our call to action is not limited to dataset scale. We call for: (1) byte-level compression metrics rather than node ratios, (2) disaggregated reporting of condensation time separately from training time, (3) cross-architecture evaluation as a standard requirement, (4) hyperparameter sensitivity analysis, and (5) theoretical guarantees. These are concrete, actionable proposals that go well beyond "use larger datasets," and we believe they collectively constitute a substantive research agenda rather than a restatement of known problems.
>
> > **Q1. The new formulation is stricter. Benefits?**
>
> The strictness is the point. Our reformulation makes three requirements explicit and enforceable that the original formulation of Jin et al. (2022b) leaves implicit:
>
> - **Byte-budget constraint** ($|\mathcal{G}_c| \leq B$): Rules out methods claiming high compression while producing large graphs, as evidenced by the 5.7× storage gap in Table 3.
> - **Generalization over model class** ($\mathcal{M}$): Rules out architecture-specific condensation when generalization is claimed, addressing the transfer failures in Tables 4 and 5.
> - **Compute efficiency** ($C\_{\text{cond}}(\mathcal{G}) + C\_{\text{train}}(\mathcal{G}\_c) \leq C\_{\text{train}}(\mathcal{G})$): Rules out methods that cost more than the problem they solve (Tables 6 and 7).
>
> > **Q2. Do smaller byte-sizes guarantee faster training time?**
>
> Not in general, and we do not claim they do. However, byte-size is a strictly better proxy than node count. In a message-passing GNN, forward pass cost is $O(|E|)$, so training time is primarily governed by edge count, not node count — a factor node-count ratio entirely obscures. More broadly, training time depends on edges ($|E|$), feature dimensionality $d$, and node count ($|V|$). Byte-size, defined as $d \cdot |V| \cdot s_f + 2 \cdot |E| \cdot s_i$, jointly accounts for all three, while also directly capturing GPU memory consumption — an equally critical constraint on resource-limited hardware.

---

> > ### Author Rebuttal · Reviewer_AmUP · 2026-04-01
> >
> > Thanks for the detailed response. I will raise score because there is really no big issue in the paper itself. But I genuinely suggest the authors to consider whether graph condensation is a valid research direction that is meaningful for real-world applications, and whether it is worth drawing more researchers into this field.
> >
> > Two more discussion if you are interested, you can just skip them if you are busy:
> >
> > For the response to W2: By mentioning the dataset condensation/distillation in the vision community, I was thinking your position paper would be better if it discuss their applications. If condensation does not really have real-world applications in vision, it it not very likely that it will work on graphs, right? Since dataset condensation itself does not bring more scalability issues in vision (since, you know, vision data comes independently so we can just do batch condensation).
> >
> > For the response to Q2: I think the correct forward cost is $O(|E|d+|V|d^2)$, not $O(|E|)$? If I understand correctly, this can also be something like $O(d\cdot\\mathrm{ByteSize})$?

---

### Official Review · Reviewer_ZJ2B · 2026-03-09

**Significance:** 2
**Argument Clarity:** 4
**Rating:** 4
**Confidence:** 3

**Questions:**

See **W1**-**W4**.

**Alternative Views Section:**

Yes

**Compliance With Llm Reviewing Policy A Conservative:**

Affirmed.

**Discussion Potential:**

3

**Paper Summary:**

This paper considered the graph condensation technique, a possible solution to handling the scalability and efficiency issues of graph learning on large-scale graphs, and comprehensively discussed limitations of related techniques. Based on such discussions, this paper further argued that current graph condensation techniques need a reset, which (1) moves beyond full-dataset training and model-dependent designs and (2) advocates lightweight, architecture-agnostic, and practically deployable approaches. It then reformulated graph condensation and provided guidelines for fair and holistic comparison.

**Position:**

Yes

**Position In Title:**

Yes

**Related Work:**

3

**Strengths And Weaknesses:**

**S1**. This paper is well-organized and well-motivated, thus easy to read.

**S2**. This paper provided a comprehensive overview of existing graph condensation techniques, especially for their common limitations.

**S3**. This paper reformulated graph condensation in a quantitative way (e.g., Definition 2.2).


***
**W1**. As claimed in this paper, graph condensation is just one of the possible solutions to applying GNNs to large-scale graphs. In addition, there exist some other solutions, e.g., node sampling, graph partitioning, low-rank embedding. From a result-oriented perspective, this paper may lack some necessary discussions about the significance of the new graph condensation techniques (discussed in this paper) beyond these alternative solutions, especially in terms of efficient and scalable graph learning.


***
**W2**. Empirical experiments in this paper are relatively simple and inconsistent, which may not fully support the major claims.

For instance, it was claimed that graph condensation can help scale a graph learning model up to large-scale graphs. However, all the datasets used in empirical experiments are classic small-scale graphs, e.g., Cora, CiteSeer, PubMed, etc. Even the largest dataset ogbn-arXiv cannot be considered as a large-scale graph. Such a limitations about dataset was discussed in Section 4 while empirical experiments still follow this limited experiment setting.

There are also no descriptions about key statistics of these datasets, e.g., numbers of nodes, edges, and classes, and feature dimensionality.

Tables 6 and 7 report results on obgn-arXiv but the corresponding results were missing in Table 5, which seem to be inconsistent.

Some details about experiment settings are missing. For instance, it is unclear how to get the mean and standard deviation values in Tables 4 and 5, e.g., over how many independent runs with what random seeds?

It seems that the discussion and empirical analysis of this paper only involves simple classic GNN models (e.g., GCN, GAT, and GIN) but lacks further discussions and evaluation for recent advances in graph transformers.


***
**W3**. This paper does not provide its code to ensure the reproducibility of experiments.


***
**W4**. There seem no discussions about possible solutions (for future research) to handling the major limitations of existing graph condensation techniques (e.g., model dependent and full-training) highlighted in this paper.

**Support:**

3

---

> ### Author Rebuttal · Authors · 2026-03-27
>
> > **W1. Include more details on alternatives to graph condensation for scalability.**
>
> We discuss this in lines 34–47 (col. 2). We propose to further expand by adding the below discussion.
>
> * **Sampling:** To further clarify the distinction: sampling can be implemented in two ways — (1) online sampling, where neighborhoods are sampled anew at each training epoch (such as in GraphSage), and (2) offline sampling, where a subgraph is sampled once a priori and training proceeds on it. It is worth noting that all graph condensation benchmarks already include the second variant as a baseline, and it has consistently been shown to be less competitive despite its efficiency. This gap exists because graph condensation pursues a dual objective: simultaneously achieving efficiency and preserving the informational content of the full graph. Sampling, by contrast, targets efficiency alone and makes no guarantees about the fidelity of the retained information. Condensation is therefore not a replacement for sampling but a strictly more ambitious goal, which, when achieved correctly, delivers strictly greater utility.
>
> * **Low-rank approximation:** Low-rank methods (e.g., sketching, spectral approximations, SVD-based compression) approximate the graph's adjacency or feature matrix with a lower-rank surrogate. However, they overlook a fundamental property of message-passing GNNs: node embeddings are a function of local neighborhood structure, meaning two nodes that are far apart in the graph may produce similar embeddings if their local neighborhoods are similar, which is a property that is independent of global rank structure. Moreover, low-rank factorization of the full adjacency matrix requires
> $O(|V|^3)$ computation, which is significantly more expensive than the $O(|E|)$ complexity of a single round of GNN message passing.
>
> * **Graph partitioning** enables distributed training across multiple computes. However, the full graph must still be stored and remains a prerequisite for partitioning itself, making it unsuitable for edge deployment. Crucially, a partition is not a standalone artifact; it cannot be transferred or reused independently of the full graph, whereas a condensed graph is a self-contained surrogate deployable without any further access to the original data.
>
> > **W2a. ..all datasets used are small-scale graphs, e.g., Cora, CiteSeer, PubMed, etc...**
>
> The reviewer may have overlooked our large-scale evaluation in **Tab. 2** on MAG240M, a graph with **240 million nodes and billions of edges**. The results are unequivocal: every gradient-based method fails entirely (OOT or OOM). This is the central empirical pillar supporting our position that techniques that rely on full-dataset training fail entirely at the scale where condensation is genuinely needed.
>
> >**W2b. no descriptions about key statistics of datasets.**
>
>  We will add the below table.
>
> |**Data**|**#Nodes**|**# Edges**|**#Classes**|**#Features**|
> |-|-|-|-|-|
> |Cora|2,708|10,556|7|1,433|
> |Citeseer|3,327|9,104|6|3,703|
> |Pubmed|19,717|88,648|3|500|
> |Flickr|89,250|899,756|7|500|
> |Ogbn-arxiv|169,343|2,315,598|40|128|
> |Reddit|232,965|23,213,838|41|602|
> |MAG240M|244,160,499|1,728,364,232|153|768|
>
> > **W2c. obgn-arXiv missing in Table 5.**
>
> Tab. 5 evaluates cross-architecture generalization. The 4 datasets included already demonstrate this failure pattern consistently. Since ours is a position paper rather than a benchmarking paper, we do not claim exhaustive dataset coverage; we claim that the evidence presented is sufficient to establish the position.
>
> >**W2d. how to get mean and standard deviation values in Tab. 4 and 5?**
>
> Every experiment has been run across 5 random seeds and we report the mean and the std dev.
>
> > **W2e. only involves classic GNN models...lacks discussions in graph transformers.**
>
> Our empirical analysis is scoped to the architectures that condensation methods themselves target, namely GCN, GAT, GIN, and GraphSAGE. These algorithms have not yet targetted transformers. Hence, establishing our position did not need evaluation on transformer.
>
> We do agree, however, that evaluation on more recent architectures (ex. graph transformers) is currently unaddressed in the field.  We will add this explicitly that the field may have other issues beyond out position, which includes evaluation on transformers.
>
> > **W3. Code.**
>
> All experiments were conducted using the official codebases released by the respective authors. Based on this feedback, we have created a repository (https://anonymous.4open.science/r/Graph-Condensation-Position-Paper) that consolidates our scripts for reproducing all reported numbers.
>
> > **W4. There seem no discussions about possible solutions...**
>
> We did not discuss any potential solutions since our primary aim is to establish our position. Nonetheless, we take this feedback in positive spirit and propose adding the discussion outlined in W2 to Rev. EBRG.

---

> > ### Author Rebuttal · Reviewer_ZJ2B · 2026-04-05
> >
> > Thanks for the rebuttal, which to some extent address most of my concerns. I decide to keep my positive score.

---

### Official Review · Reviewer_wcbJ · 2026-03-13

**Significance:** 3
**Argument Clarity:** 3
**Rating:** 5
**Confidence:** 4

**Questions:**

1. If the end goal is to perform well on classification tasks, how would condensation method compared with simple sampling methods like GraphSAGE? Is condensation actually needed for the field?

**Alternative Views Section:**

Yes

**Compliance With Llm Reviewing Policy A Conservative:**

Affirmed.

**Discussion Potential:**

3

**Paper Summary:**

This paper advocates for a refresher on the graph condensation research, where misleading evaluation metrics and impracticality is the norm. Graph condensation aims to produce a more compact graph given a very large graph input such that a graph neural network trained on the smaller graph would perform similarly with one trained on the large graph. The paper redefined this graph condensation problem and standardized key evaluation dimensions for the desired properties. The paper performed a comprehensive comparison of existing graph condensation methods as well as demonstrating the inability to scale for most of the condensation methods. Experimental results further supported their concerns on comparison metrics in the literature. The paper proposed a principled steps toward fair comparison and theoretical foundations for graph condensation research.

**Position:**

Yes

**Position In Title:**

Yes

**Related Work:**

4

**Strengths And Weaknesses:**

# Strength

- The paper addressed a valid issue in a modern and important research direction.
- The paper's writing is clear, with thorough citations and complete experimental data to support the position.
- The position proposed could ignite strong discussion and realign the community toward higher quality research

# Weakness

**Support:**

3

---

> ### Author Rebuttal · Authors · 2026-03-26
>
> We thank Reviewer wcbJ for the thorough and encouraging review, and for the positive assessment of the paper's contributions. We are glad the position resonated as both timely and well-supported. The reviewer raises one question, which we address below.
>
> >**Q1. If the end goal is to perform well on classification tasks, how would condensation method compared with simple sampling methods like GraphSAGE? Is condensation actually needed for the field?**
>
> This is an important question, which we discuss directly in lines 34–47 (column 2). We reproduce the relevant passage below for reference and then expand on it.
>
> > Scalability techniques such as node sampling\cite{graphsage, fastgcn}, graph partitioning\cite{graph_partition, clustergcn, zeng2019graphsaint}, and low-rank embedding\cite{sketchgnn, low_rank} aim to reduce resource demands, but they often struggle to simultaneously preserve global structural information, minimize dataset size, and ensure efficient computation. Sampling-based training, though efficient in theory, can become prohibitively expensive at scale, consuming up to 62.7\% of runtime as graph size increases \cite{liu2022gnnsamplerbridginggapsampling}. Graph condensation~\citep{gcond,doscond,msgc,sgdd,exgc,geom,mcond} offers a more promising direction: it seeks to synthesize a smaller graph $G\_c=(V\_c, E\_c, F\_c)$ such that a model trained on $G\_c$ achieves predictive performance comparable to that of a model trained on the original graph $G = (V, E, F)$.
>
> To further clarify the distinction: sampling can be implemented in two ways — (1) online sampling, where neighborhoods are sampled anew at each training epoch (such as in GraphSage), and (2) offline sampling, where a subgraph is sampled once a priori and training proceeds on it. It is worth noting that all graph condensation benchmarks already include the second variant as a baseline, and it has consistently been shown to be less competitive despite its efficiency. This gap exists because graph condensation pursues a dual objective: simultaneously achieving efficiency and preserving the informational content of the full graph. Sampling, by contrast, targets efficiency alone and makes no guarantees about the fidelity of the retained information. Condensation is therefore not a replacement for sampling but a strictly more ambitious goal, which, when achieved correctly, delivers strictly greater utility.

---

> > ### Author Rebuttal · Reviewer_wcbJ · 2026-04-06
> >
> > I thank the authors for their detailed rebuttal. My concern was addressed and I maintained my accept decision.

---

### Official Review · Reviewer_EBRG · 2026-03-15

**Significance:** 4
**Argument Clarity:** 3
**Rating:** 5
**Confidence:** 4

**Questions:**

Please see the Weaknesses

**Alternative Views Section:**

Yes

**Compliance With Llm Reviewing Policy A Conservative:**

Affirmed.

**Discussion Potential:**

3

**Paper Summary:**

This paper argues that current definition and evaluation pipeline of graph condensation need to be reset. Specifically, the current full-dataset training paradigm and gradient matching-based solution may cause high computational overhead, which obeys the basic objective of condensation. On the other hand, current small-size datasets in the benchmark also misleads the practical usefulness of condensation algorithm. The authors claim that the community should moving the focus to lightweight, model-agnostic, and practically deployable graph condensation methods.

**Position:**

Yes

**Position In Title:**

Yes

**Related Work:**

3

**Strengths And Weaknesses:**

Strengths:
1. The proposed viewpoints are insightful and can inspire the current research.
2. The paper includes a comprehensive summarization and discussion for the existing studies.
3. Experiments are conducted to support the viewpoints in this paper.

Weaknesses:
1. The critique of gradient-based methods appears somewhat one-sided, and the discussion of their practical benefits is relatively limited. Meanwhile, there are also some condensation methods beyond gradient-based, which needs more discussions.
2. The proposed reformulation of graph condensation is largely theoretical and abstract, and the paper does not discuss how such a formulation could be optimized or implemented in practice.

**Support:**

3

---

> ### Author Rebuttal · Authors · 2026-03-26
>
> We thank Reviewer EBRG for the careful and constructive reading of our submission. Please find below our clarifications.
>
> > **W1a. The critique of gradient-based methods appears somewhat one-sided, and the discussion of their practical benefits is relatively limited.**
>
> We would like to gently clarify that **Section 5 (Alternative Views and Counter-Perspectives)** is dedicated to acknowledging the genuine practical merits of gradient-based methods. We present three scenarios in which they retain legitimate value:
>
> - **Architecture-specific production pipelines:** In many deployment settings, a single GNN architecture is standardized. A condensed graph tuned to that architecture may be entirely acceptable — and even preferable — to a general-purpose one that trades off peak performance for portability.
> - **One-time amortized computation:** Large organizations can absorb the high upfront condensation cost by distributing the resulting synthetic graph internally, accelerating repeated hyperparameter sweeps without re-running full-graph training.
> - **Rich training signal extraction:** Gradient trajectories encode fine-grained loss landscape information that simpler distribution-matching approaches may miss, particularly on challenging datasets such as ZINC and PATTERN where repeated noise filtering via gradient-based methods can be advantageous.
>
> We hope this clarifies that our position is not a blanket dismissal of gradient-based methods, but a targeted argument that they are misapplied when scalability and architecture-agnosticism are the stated goals.
>
> If the reviewer feels, we would be happy to expand this discussion further.
>
> > **W1b. Meanwhile, there are also some condensation methods beyond gradient-based, which needs more discussions.**
>
> We humbly point out that **Section 3.2** is entirely dedicated to discuss model-agnostic methods that are not gradient-based, covering BONSAI, MIRAGE, TMD, GCDM, DISCO, GCPA, GDEM, BIMSGC, CTGC, and CLUSTGDD, including their design principles, strengths, and residual limitations.
>
> In addition, **Table 1** organizes 33 methods across a two-axis taxonomy of model-dependence and full-dataset training requirement, directly capturing the landscape beyond gradient-matching.
>
> We are happy to expand any specific discussion that the reviewer feels is insufficient.
>
> >**W2. The proposed reformulation of graph condensation is largely theoretical and abstract, and the paper does not discuss how such a formulation could be optimized or implemented in practice.**
>
> This design choice to keep the formulation algorithm-agnostic was made to cater to the objectives of a Position paper as discussions on potential solutions or implementations would dilute the separation with a regular research track paper.
>
> We do agree, however, that explicitly connecting the formal definition to the algorithmic design space would strengthen the paper, and we propose to add the following paragraph to Sec. 2 in our revision.
>
> > The four necessary conditions in Def.2.2 carry direct implications for algorithm design.
> > * **Generalization across model class M** rules out any method that derives its condensation objective from architecture-specific signals, including gradient matching, neural tangent kernels, and attention-map alignment, and instead favors methods grounded in architecture-invariant graph statistics such as degree distributions, Weisfeiler-Lehman subtree frequencies, or feature-space cluster summaries.
> >* **Compactness under a byte budget** guards against synthesis-based methods
>   that condense a large sparse graph into a smaller but denser one — a common
>   failure mode that may yield little to no reduction in memory consumption or
>   training time, since forward pass complexity is $O(|E|)$ and a
>   denser condensed graph can offset the savings from fewer nodes. It also
>   ensures fair comparison across methods, as illustrated by the 5.7× storage
>   gap in Table 3 between methods all claiming "1% condensation." For
>   learning-based methods, this constraint could be operationalized via a
>   regularizer that jointly penalizes edge density and feature dimensionality in
>   the condensed graph — for instance, an $L_1$ penalty on the adjacency
>   weights to encourage sparsity, and a low-rank or sparsity-inducing penalty on
>   the synthetic node features.
>
> > * **Compute efficiency:** Since condensation must cost strictly less than full-graph training, it points toward single-pass or embarrassingly parallel algorithms.
>
> > * **$\epsilon$-Task preservation** is the hardest property to guarantee formally. A natural direction is to exploit the Lipschitz continuity of popular graph neural networks [1]. If the condensed graph introduces bounded distortion relative to the original, Lipschitz continuity provides a principled pathway to bounding the resulting performance gap $\epsilon$.
>
> [1] Juvina et al., Training Graph Neural Networks Subject to a Tight Lipschitz Constraint, TMLR 2024.

---

> > ### Author Rebuttal · Reviewer_EBRG · 2026-04-03
> >
> > I appreciate the authors for the reply. I'll keep my positive score.

---

### Decision · Program_Chairs · 2026-04-30

**Decision:**

Accept (spotlight)

**Comment:**

All reviewers agreed that this is an important contribution that should be accepted as a position paper.